# Assessment of Cardio-Respiratory Function in Overweight and Obese Children Wearing Face Masks during the COVID-19 Pandemic

**DOI:** 10.3390/children9071053

**Published:** 2022-07-14

**Authors:** Riccardo Lubrano, Silvia Bloise, Mariateresa Sanseviero, Alessia Marcellino, Claudia Proietti Ciolli, Enrica De Luca, Alessia Testa, Anna Dilillo, Saverio Mallardo, Sara Isoldi, Vanessa Martucci, Emanuela Del Giudice, Rita Leone, Donatella Iorfida, Flavia Ventriglia

**Affiliations:** Dipartimento Materno Infantile e di Scienze Urologiche, Sapienza Università di Roma, UOC di Pediatria e Neonatologia-Polo Pontino, 04100 Latina, Italy; riccardo.lubrano@uniroma1.it (R.L.); mariateresa.sanseviero@yahoo.it (M.S.); marcellino.alessia@gmail.com (A.M.); proietticiolli.claudia@gmail.com (C.P.C.); enrideluca@live.it (E.D.L.); alessiatesta92@live.it (A.T.); annadilillo83@gmail.com (A.D.); saverio.mallardo@gmail.com (S.M.); isoldi.sara@gmail.com (S.I.); vany.mart@gmail.com (V.M.); emanuela.delgiudice@gmail.com (E.D.G.); mrita.leone@yahoo.it (R.L.); donatella.iorfida@gmail.com (D.I.); flavia.ventriglia@uniroma1.it (F.V.)

**Keywords:** children, obesity, COVID-19, respiratory function, prevention

## Abstract

Objective: To evaluate whether the use of a surgical and N95 mask for overweight and obese children was associated with respiratory distress. Methods: We enrolled 15 healthy and 14 overweight or obese children. We performed two sessions: one wearing a surgical, the other an N95 mask. We tracked changes in partial pressure of end-tidal carbon dioxide (PETCO2), oxygen saturation (SaO2), pulse rate (PR), and respiratory rate (RR) during a 72 min test: 30 min without a mask, 30 min wearing a mask, and then during a 12 min walking test. Results: In healthy children, there was no significant change in SaO2 and PETCO2 during the study; there was a significant increase in PR and RR after the walking test with both the masks. In overweight or obese children, there was no significant change in SaO2 during the study period; there was a significant increase in PETCO2 as fast as wearing the mask and an increase in PETCO2, PR, and RR after walking test. After the walking test, we showed a significant correlation between PETCO2 and body mass index. Conclusion: Overweight or Obese children who wear a mask are more prone to developing respiratory distress, which causes them to remove it frequently. In a crowded environment, they are at greater risk of infection. For this reason, it is desirable that they attend environments where everyone uses a mask.

## 1. Introduction

Individuals with obesity are among the groups most at risk of SARS-CoV-2 infection-related morbidity and mortality [1,2], because of an impaired baseline pulmonary function [3,4,5] and reduced immune function [6,7].

Although coronavirus disease 2019 (COVID-19) has a milder course in children [8,9] recent reports showed that obesity is one of the most prevalent comorbidities among severe cases of SARS-CoV-2 infection in childhood and adolescence [10,11].

According to the U.S. Centers for Disease Prevention and Control (CDC) [12] and the American Academy of Pediatrics (AAP) [13] in addition to social distance and handwashing, the use of face masks is very important for children, reducing the possibility of the spread of SARS-CoV-2.

This recommendation on mask use in children is a key point, considering both that many children are unvaccinated, and the frequent asymptomatic course of COVID-19 is in the pediatric age group.

For this reason, it would be appropriate for all children to use a face mask when attending crowded environments. In fact, new CDC research conducted in 1000 schools in Arizona’s Maricopa and Pima counties showed that schools without mask requirements were three and a half times more likely to have COVID-19 outbreaks than those that implemented the mask requirement, respectively, with 113 COVID-19 outbreaks in the former, versus 16 outbreaks in the latter in the first month of in-person learning [14].

Despite the proven effectiveness of this preventive strategy [15,16,17,18] the topic of the use of face masks has been very debated; in fact, many concerns have arisen about the possible side effects on cardiopulmonary function related to the use of masks [19,20] in healthy children and especially in those with cardiac and/or respiratory diseases, even in the absence of sufficient scientific evidence. Only recently, some reports have evaluated the effects on respiratory function of wearing a mask in children and adult people [21,22,23].

Nevertheless, to date, there are no studies on children with obesity, which, however, have all the potential to be considered a high-risk category.

Therefore, given the importance of wearing protective masks in children with obesity, we wanted to assess whether the use of a surgical or N95 mask could be dangerous, favoring the occurrence of episodes of desaturation or respiratory distress.

## 2. Materials and Methods

### 2.1. Trial Design

This was a non-randomized controlled clinical trial. This study was conducted at the pediatric department of the Goretti Hospital, Polo Pontino Sapienza University of Rome. The study was approved by the Institutional Review Board of the Maternal and Child Health Department of the Local Health Authority of Latina (protocol 02-15/03/2021). This report followed the CONSORT reporting guidelines for clinical trials. For each child, verbal informed consent was obtained from both parents and the study protocol conformed to the ethical guidelines of the 1975 Declaration of Helsinki as revised in 2000 [24].

### 2.2. Participants and Study Period

From June 2021 to July 2021, we enrolled 15 healthy children (group HC) and 14 children with a diagnosis of overweight and obesity (group OC).

Inclusion criteria for group HC were: patients with a normal weight, without any comorbidities, and not taking any type of medication.

Inclusion criteria for group OC were: patients with a diagnosis of overweight and obesity, without lung or cardiac diseases, and who were not taking medications influencing the parameters considered. The diagnosis of overweight and obesity was based on body mass index (BMI) according to the WHO 2006 reference tables for ages 2–5 years [25,26] the WHO 2007 reference tables [27] for ages 5–18 years, or the SIEDP 2006 reference tables [28] for ages 2–18 years.

### 2.3. Interventions

For each patient enrolled we performed 2 tests, one wearing a surgical mask, and the other wearing an N95 mask. During both tests, all subjects were monitored every 30 min, the first 30 min while not wearing a mask (T30), and the next 30 min while wearing a mask (T60), and then performed a 12 min walking test (Twt). The instructions given for this last session were similar to those for the 12 min walking test (wt) [29]. During the study, every child was connected to a Masimo Patient Monitoring System (Rad-97™ with NomoLine Capnography, Neuchâtel Switzerland) to log partial pressure of end-tidal carbon dioxide (mmHg), oxygen saturation (%), and pulse rate (pulsation/min). The respiratory rate (breaths/min) was measured manually by an observer. All parameters were recorded every 30 min at 30 (T30) and 60 min (T60) from the start of the test, and a final assessment at the end of the 12 min walking test (Twt). The supervising doctor was appointed to identify any signs of respiratory distress, including the use of accessory muscles of respiration and onset of pallor or cyanosis. The masks used for the tests were: single-use surgical masks (BYD Precision Manufacture, Changzhou, China) and N95 masks (Dongguan AOXING AV Equipment Co., Ltd., Dongguan, China). Before the study started, for every patient enrolled, we implemented the following steps: the physical examination to check the child’s state of well-being; the fit check test (user leak check, self-check) to verify the facial tightness by the absence of air leaks using positive and negative pressure checks [30].

### 2.4. Objectives

The primary aim was to evaluate whether the use of surgical or N95 masks among overweight or obese children is associated with episodes of oxygen desaturation or respiratory distress and whether there were any differences with healthy children.

The secondary aim was to assess whether in children overweight or obese there was a correlation between the body mass index (BMI) and the risk of developing respiratory distress.

### 2.5. Main Outcomes and Measures

The main outcomes of this study were the changes in respiratory parameters during the use of the masks: partial pressure of end-tidal carbon dioxide (PETCO2), oxygen saturation (SaO2), pulse rate (PR) respiratory rate (RR); and the presence of clinical signs of respiratory distress.

### 2.6. Statistical Analysis

For statistical analysis, we relied on JMP 16.1.0 program for Mac by SAS Institute inc. For the data expressed as continuous variables the approximation to normal of the distribution of the population was tested with the Shapiro–Wilk and Anderson Darling test. As results were asymmetrically distributed, data are expressed as the median and interquartile range (IQR), 25th and 75th quartile, and non-parametric tests were used. Data were analyzed with the Kruskal–Wallis nonparametric one-way analysis of variance to examine the changes of parameters in both groups on T30, T60, and Twt. The null hypothesis was that the groups of the study all came from the same distribution. When the Kruskal–Wallis test was significant, we used the Wilcoxon test to compare the intragroup differences at the five observation times. The nonparametric correlation between two variables has been evaluated with Spearman’s Rho (r) correlation. A *p* < 0.05 was considered significant.

## 3. Results

A total of 29 patients were enrolled, 12 males and 17 females. Of these, 15 were children with normal weight (group HC) and 14 were obese or overweight children (group OC) (Table 1); 3 healthy children refused to perform the test with surgical masks while agreeing to wear an N95 mask.

Of course, we can observe that the BMI of the two groups is significantly different: healthy children 18.61 (18.08–20.83) children with obesity 33.75 (24.43–38.85) (*p* < 0.0001).

Table 2 shows the analysis of the readings of SaO2 (%), PETCO2 (mmHg), PR (pulsation/min), RR (breaths/min), and the occurrence of clinical signs of respiratory distress at T30, T60, and after walking test (Twt) in the group of healthy children wearing first the surgical mask and then the N95 mask.

In the group HC, there was no statistically significant change in SaO2 and PETCO2 both wearing a surgical mask and wearing an N95 mask during the study period. Comparing the values of PETCO2 between HC wearing surgical masks and HC wearing an N95 masks, we showed that there were no statistically significant changes at T30 and T60 (*p* = NS), while there was an increase in PETCO2 wearing an N95 mask after the walking test (*p* = 0.0088).

After the walking test (Twt), we showed a significant increase in PR and RR both wearing a surgical mask and both N95 masks compared to T30 and T60, while there were no differences in PR and RR between T30 and T60.

At the end of the walking test, all health children did not show clinical signs of respiratory distress: tachypnea, retractions, and dyspnea.

The average distance travelled by HC during the walking test was similar whether wearing the surgical mask or the N95.

[SM: 760 m (555–800); N95: 760 m (555–871); *p*: NS].

Table 3 shows the analysis of the readings of SaO2 (%), PETCO2 (mmHg), PR (pulsation/min), RR (breaths/min) and the occurrence of clinical signs of respiratory distress at T30, T60 and after walking test (Twt) in the group of children with obesity wearing first the surgical mask and then the N95 mask.

In the group OC, there was no statistically significant reduction in SaO2 during the study period, while there was a significative increase in PETCO2 between T30 and T60 and after the walking test both in children wearing a surgical mask [Kruskal–Wallis *p* < 0.0001; T30 vs. T60 *p* = *0*.0007; T30 vs. Twt *p* = 0.0004; T60 vs. Twt *p* = 0.0456] and in children wearing N95 mask [Kruskal–Wallis *p* < 0.0001; T30 vs. T60 *p* = 0.0108; T30 vs. Twt *p* = 0.0009; T60 vs. Twt *p* = 0.0343].

Comparing the values of PETCO2 between OC wearing a surgical mask and OC wearing an N95 mask there were no statistically significant changes.

After the walking test, we showed a significant increase in PR and RR both wearing a surgical mask both N95 masks compared to T30 and to T60, while there were no differences in PR and RR between T30 and T60.

At the end of the walking test, all obese children showed clinical signs of respiratory distress: tachypnea, retractions and dyspnea (Table 3).

The average distance traveled by OC during the walking test was similar whether wearing the surgical mask or the N95. [SM: 765 m (687.5–825); N95: 760 m (555–871); *p*: NS].

In addition, we analyzed the nonparametric correlation between PETCO2 and BMI when children with obesity wore the surgical mask or the N95 mask at T60 and at Twt.

At T60, we did not find a significant nonparametric correlation between PETCO2 and BMI (with SM: Spearman Rho = 0.3036 *p* = 0.2913; with N95 Spearman Rho = 0.4798 *p* = 0.0825); While, after walking test, with both masks, we found a nonparametric significative correlation between PETCO2 and BMI (with SM: Spearman Rho = 0.5503 *p* = 0.0414; with N95 Spearman Rho = 0.7381 *p* = 0.026) (Figure 1).

The above figure is a comparison between the group of healthy children and the group of overweight and obese children. 

For the first step with the surgical mask, we found no significant difference between the group of healthy children and the group of overweight and obese children at T30, while we found a significant increase in P_ETCO2_ at T60 (*p* = 0.0046) and a significant increase in P_ETCO2_ (*p* = 0.0001) and respiratory rate (*p* = 0.041) at Twt in the group of overweight and obese children compared with healthy controls (Table 4); for the second step with the N95 mask, we found no significant difference between the two groups at T30 and T60, while we found a significant increase in P_ETCO2_ (*p* = 0.0062), pulse rate (*p* = 0.01) and respiratory rate (*p* = 0.0053) at Twt in the group of overweight and obese children compared with healthy controls (Table 5). In addition, the group of obese and overweight children showed clinical signs of respiratory distress (tachypnea, retractions, and dyspnea) at the end of the walking test with both the surgical mask and the N95 mask.

## 4. Discussion

Our study showed that overweight or obese children wearing surgical or N95 masks are at risk of experiencing a change in their cardio-respiratory function, evidenced by an early increase in PETCO2 as soon as the children wore a mask, and to a greater extent after the walking test, and also associated after exercise with an increase in respiratory and heart rate and the appearance of clinical signs of respiratory distress.

This could be to the fact that patients with obesity have an accumulation of adipose tissue in the abdomen and thorax that causes a reduction in lung volume and pulmonary compliance with consequent alteration of the breathing pattern. This results in a reduction in functional residual capacity (FRC) and expiratory reserve volume with the consequent risk of expiratory flow limitation, airway closure, and abnormalities in ventilation distribution [3,31,32]. Furthermore, another aspect to consider is the increase in anaerobic metabolism in people with obesity, as suggested by different reports that showed reduced adipose tissue oxygenation [33,34,35]. Related to this, Balmain et al. [33] showed in an obese subject that there is a decreased V E/V CO2 (minute ventilation/carbon dioxide elimination) slope after a physical effort that reflects a blunted ventilatory response with increasing BMI. This could be due to CO2 retention that occurs in these subjects during exercise as a result of mechanical constraints on breathing related to the typical changes seen in obesity. This also explains why in subjects with obesity, unlike normal children, there is a further increase in PETCO2 after the walking test, associated with clinical signs of respiratory distress [36]. In fact, in agreement with Balman et al. [36], also in our study we showed a positive, nonparametric correlation between PETCO2 and BMI only after the walking test.

About this, we assume that the increase in PETCO2 in overweight or obese children, 60 min after wearing the surgical mask or N95 mask at rest, could be the result of the combination of mechanical factors (accumulation of adipose tissue that alters the lung volumes) and metabolic factors (natural tendency to have reduced oxygenation of the adipose tissue) typical of the subjects with obesity, worsened by the presence of a face mask, which determined an increase in the dead space volume and of resistance to airflow 23 causing further impairment of respiratory dynamics and respiration of CO2, although it has been shown that the obstruction to airflow is below the National Institute for Occupational Safety and Health (national and Roberge) safety limits [37,38].

The result is that the combination of the mask effect, with the altered ventilatory mechanics and the tendency to anaerobic metabolism, are further aggravated after the walking test and is characterized by a further increase in PETCO2 and an increase in respiratory and cardiac rate associated with the development of clinical signs of respiratory distress.

In contrast, these findings do not occur when we performed the tests on healthy children with both types of masks.

Despite the development of this clinical and metabolic condition, overweight or obese children in our study did not experience episodes of desaturation; however, the increase in PETCO2 should be considered very carefully, because it is an early sign of alveolar hypoventilation [39,40,41,42,43] preceding hypoxia.

In line with this evidence, we suggest that people with obesity can wear a mask without problems at rest and for limited time periods, but we believe that should not wear a mask during physical activity or for a prolonged time, avoiding areas of substantial or high transmission.

Furthermore, we believe that overweight or obese children could be more easily induced to remove the mask because of the potential occurrence of respiratory distress, thus increasing the risk of contracting SARS-CoV-2 infection.

Therefore, to ensure maximum safety for children with obesity, they should frequent environments in which the general population presents a high degree of mask-use compliance in order to obtain maximum protection effectiveness and “preserve” the most vulnerable people.

This study has some limitations: firstly, small sample size; secondarily, we did not perform spirometry or lung volume measurements in our study group.

## Figures and Tables

**Figure 1 children-09-01053-f001:**
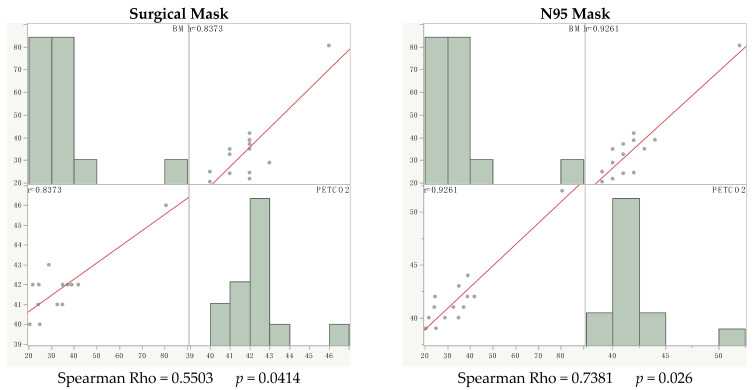
Spearman’s Rho (r) correlation between the degree of obesity (BMI) and the levels P_ETCO2_ after the walking test wearing surgical mask (SM) and N95 mask (N95).

**Table 1 children-09-01053-t001:** Demographic data of enrolled children.

Children	Age	Weight	Height	Body Mass Index	Males/Females
	Months	kg	cm		*n*/*n*
**Healthy**	108	39	139	18.61	8/7
	72–130	28.8–42	131–145	18.08–20.83	
**Obese**	144	73.95	149	33.75	4/10
	107.5–184.75	48.55–92.45	139–157.88	24.43–38.85	

**Table 2 children-09-01053-t002:** Oxygen saturation (SaO2), partial pressure of end-tidal carbon dioxide (P_ETCO2_), pulse rate (PR), respiratory rate (RR), and the occurrence of clinical signs of respiratory distress during the test, at 30′ (T30), at 60′ (T60) minutes and after a 12 min walking test (Twt) in all healthy children wearing a surgical mask (SM) and N95 mask (N95). All data are expressed as median, 25th, and 75th quartile.

	Test with Surgical Mask				Test with N95 Mask			
	Without Mask	With Mask	With Mask and Walking Test	Kruskal–Wallis Test	Without Mask	With Mask	With Mask and Walking Test	Kruskal–Wallis Test
	T30	T60	Twt		T30	T60	Twt	*p*
**SaO2**	98	98	98	NS	98	98	97	NS
%	97–98	97–98	96–98		97–98	96–98	96–98	
**P_ETCO2_**	36	36	35	NS	35	38	39	NS
mmHg	35–37	33–37	33–39		35–40	35–41	37–41	
**PR**	96	106	115	**<0.0015**	99	100	109	**<0.0304**
pulsation/min	89–100	100–115	110–120		87–105	82–106	105–114	
**RR**	22	21	26	**<0.0001**	21	22	25	**<0.0098**
breaths/min	20–23	21–22	24–28		19–22	20–23	20–30	
**Clinical signs of respiratory distress ***	Absent	Absent	Absent		Absent	Absent	Absent	

* Clinical signs of respiratory distress: tachypnea, retractions, and dyspnea.

**Table 3 children-09-01053-t003:** Oxygen saturation (SaO2), partial pressure of end-tidal carbon dioxide (P_ETCO2_), pulse rate (PR), respiratory rate (RR) and the occurrence of clinical signs of respiratory distress during the test, at 30′ (T30), 60′ (T60) minutes and after a 12 min walking test (Twt) in all overweight and obese patients wearing a surgical mask (SM) and N95 mask (N95). All data were expressed as median, 25th and 75th quartile.

	Test with Surgical Mask				Test with N95 Mask			
	Without Mask	With Mask	With Mask and Walking Test	Kruskal–Wallis Test	Without Mask	With Mask	With Mask and Walking Test	Kruskal–Wallis Test
	T30	T60	Twt		T30	T60	Twt	*p*
**SaO2**	98	97.5	97.5	NS	98	98	97	NS
%	96.75–98.25	96.75–98	96–98		96.75–98.25	97–98	96–98	
**P_ETCO2_**	37	39	42	**<0.0001**	37	39	41.5	**<0.0001**
mmHg	35.75–39	38–43.25	41–42		35.75–39.25	38.75–41.75	40–43.25	
**PR**	93	90.5	127	**<0.0001**	93	94	131	**<0.0002**
pulsation/min	85.75–100.5	85.25–102.25	118.75–137.25		85.75–100.5	88.25–105.25	112.25–140	
**RR**	20	20	30	**<0.0001**	20	20.5	32	**<0.0001**
breaths/min	16–21.25	16–23	25.5–36		16–21.25	18–23.25	25.5–34	
**Clinical signs of respiratory distress** *	Absent	Absent	**Present**		Absent	Absent	**Present**	

* Clinical signs of respiratory distress: tachypnea, retractions, and dyspnea.

**Table 4 children-09-01053-t004:** Comparison between the group of healthy children and the group of overweight and obese children wearing a surgical mask. All data were expressed as median, 25th, and 75th quartile.

Test with Surgical Mask									
	Without Mask			With Mask			With Mask and Walking Test		
	T30			T60			Twt		
	Healthy	Obese	*p*	Healthy	Obese	*p*	Healthy	Obese	*p*
**SaO2**	98	98	NS	98	97.5	NS	98	97.5	NS
%	97–98	96.75–98.25		97–98	96.75–98		96–98	96–98	
**P_ETCO2_**	36	37	NS	36	39	**0.0046**	35	42	**0.0001**
mmHg	35–37	35.75–39		33–37	38–43.25		33–39	41–42	
**PR**	96	93	NS	106	90.5	NS	115	127	NS
pulsation/min	89–100	85.75–100.5		100–115	85.25–102.25		110–120	118.75–137.25	
**RR**	22	20	NS	21	20	NS	26	30	**0.041**
breaths/min	20–23	16–21.25		21–22	16–23		24–28	25.5–36	
**Clinical signs of respiratory distress ***	Absent	Absent		Absent	Absent		Absent	**Present**	

***** Clinical signs of respiratory distress: tachypnea, retractions, and dyspnea.

**Table 5 children-09-01053-t005:** Comparison between the group of healthy children and the group of overweight and obese children wearing an N95 mask. All data were expressed as median, 25th, and 75th quartile.

Test with surgical mask									
	Without Mask			With Mask			With Mask and Walking Test		
	T30			T60			Twt		
	Healthy	Obese	*p*	Healthy	Obese	*p*	Healthy	Obese	*p*
**SaO2**	98	98	NS	98	98	NS	97	97	NS
%	97–98	96.75–98.25		96–98	97–98		96–98	96–98	
**P_ETCO2_**	35	37	NS	38	39	NS	39	41.5	**0.0062**
mmHg	35–40	35.75–39.25		35–41	38.75–41.75		37–41	40–43.25	
**PR**	99	93	NS	100	94	NS	109	131	**0.01**
pulsation/min	87–105	85.75–100.5		82–106	88.25–105.25		105–114	115.25–140	
**RR**	21	20	NS	22	20.5	NS	25	32	**0.0053**
breaths/min	19–22	16–21.25		20–23	18–23.5		20–30	25.5–34	
**Clinical signs of respiratory distress ***	Absent	Absent		Absent	Absent		Absent	**Present**	

***** Clinical signs of respiratory distress: tachypnea, retractions, and dyspnea.

## Data Availability

All data and materials support published claims and complied with field standards.

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
