# Peer review of "Assessment of Cardio-Respiratory Function in Overweight and Obese Children Wearing Face Masks during the COVID-19 Pandemic"

_children, 2022, doi:10.3390/children9071053_

Round 1

Reviewer 1 Report

Authors Made changes according to the review requestes

Author Response

Authors Made changes according to the review requestes

Dear Reviewer,

Thank you for your consideration of our manuscript. We truly think the manuscript is improved after the revisions suggested.

Reviewer 2 Report

Thank you for this clarification.

There remain some minor editorial things to be improved:

Lines 134/ 135 Table 1: children not patients and title should be on the same page as the table.

Lines 186- 189 redundant, delete

Lines 215-225: normal text

Line 234: Table 5:  The whole table should be on one page

General: all titles of the tables should be either normal or bold.

Author Response

Thank you for this clarification.There remain some minor editorial things to be improved:

Dear Reviewer,

Thank you for your consideration of our manuscript. We truly think the manuscript is improved after the revisions suggested.

We now submit our revised manuscript for publication in Children. We marked the revisions through the manuscript using the “track changes”.

Lines 134/ 135 Table 1: children not patients and title should be on the same page as the table.

We have now modified the patients with children and moved the title to the same page as table 1.

Lines 186- 189 redundant, delete

We have now deleted the lines 186-189.

Lines 215-225: normal text

Now it is normal text

Line 234: Table 5:  The whole table should be on one page

Now, the whole table 5 is on one page

General: all titles of the tables should be either normal or bold

Now all titles of tables are in bold.

This manuscript is a resubmission of an earlier submission. The following is a list of the peer review reports and author responses from that submission.

Round 1

Reviewer 1 Report

Unfortunately, the main results are not presented in tables which makes the manuscript impossible to judge.

Author Response

Dear Editors and Reviewers,

Thank you for your consideration of our manuscript. We truly think the manuscript is improved after the revisions suggested. Below we respond in detail to the comments and points the reviewer raised. We now submit our revised manuscript for publication in Children. We marked the revisions through the manuscript using the “track changes”.

Reviewer 1

Unfortunately, the main results are not presented in tables which makes the manuscript impossible to judge.

We thank the reviewer for his suggestion. In table 2 and in table 3 are shown the analysis of the readings of SaO2 (%), PETCO2 (mmHg), PR (pulsation/min), RR (breaths/min) and the occurrence of clinical signs of respiratory distress at T30, T60 and after walking test (Twt) in the group of healthy children and in the group of children with obesity wearing first the surgical mask and then the N95 mask. Furthermore, we have now added another paragraph in the results section where the results of the comparison between the two groups are expressed. Specifically, for the first step with the surgical mask, we found no significant difference between the two groups at T30 , while we found a significant increase in PETCO2 at T60 ( p=0.0046) and a significant increase in PETCO2 ( p= 0.0001) and respiratory rate ( p= 0.041) at Twt in the group of obese children compared with healthy controls; for the second step with the N95 mask, we found no significant difference between the two groups at T30  and T60, while we found a significant increase in PETCO2 (p=0.0062), pulse rate ( p= 0.01) and respiratory rate ( p= 0.0053) at Twt in the group of obese children compared with healthy controls.

Reviewer 2 Report

Thank you for giving me the opportunity to read the paper.

Below, my observations:

English must be improved 

There are some ortographic errors

The paper tells about 29 patients, 14 with obesity or overweight (OC group). The title of the paper is "assessment of respiratory function in obese children wearing face masks....", I would change the title in "assessment of respiratory function in overweight and obese children wearing face masks...".

The same observation should be taken in account every time the authors referer to obese children in the study from the abstract to the conclusions.

In both table 1 and 2 it would be appropriate to indicate the significant statistical differences. This make it easier to read the results and understand the meaning of the study

The authors talk about respiratory function. I agree when they talk about respiratory rate. How does the heart rate fit into "respiratory function" context? The SaO2/CO2 monitoring are expression of gas exchanges rather than respiratory function. I would suggest to consider the change from respiratory function to cardiorespiratory parameters or similar.

Why wasn't spirometry or lung volumes measurement considered? I would suggest to include in the discussion a short "Limitations of the study" paragraph.

In the abstract, I would change the sentence "obese children who wear a mask are at risk of developing respiratory distress...." as no statistical analysis on risk has been performed, I would suggest to use "are more prone of developing ... (or similar)"

Author Response

Dear Editors and Reviewers,

Thank you for your consideration of our manuscript. We truly think the manuscript is improved after the revisions suggested. Below we respond in detail to the comments and points the reviewer raised. We now submit our revised manuscript for publication in Children. We marked the revisions through the manuscript using the “track changes”.

Reviewer 2

Below, my observations:

English must be improved 

We have now revised English language to improve fluency and quality of the paper and corrected spelling and grammatical errors throughout the text.

There are some ortographic errors

We have now corrected ortographic errors

The paper tells about 29 patients, 14 with obesity or overweight (OC group). The title of the paper is "assessment of respiratory function in obese children wearing face masks....", I would change the title in "assessment of respiratory function in overweight and obese children wearing face masks...".

Yes, we agree with the reviewer, we have now modified the title in “assessment of cardio-respiratory function in overweight and obese children wearing face masks during the Covid-19 pandemic".

The same observation should be taken in account every time the authors referer to obese children in the study from the abstract to the conclusions.

Yes, we have now changed the terminology, using in overweight and obese children throughout the text

In both table 1 and 2 it would be appropriate to indicate the significant statistical differences. This make it easier to read the results and understand the meaning of the study

In both tables, we have highlighted statistically significant differences in bold type to make the results clearer and easier for the reader to understand. In addition, to make the results clearer we have added a paragraph in the results section, in which we describe the differences between the two study groups. Specifically, for the first step with the surgical mask, we found no significant difference between the two groups at T30 , while we found a significant increase in PETCO2 at T60 ( p=0.0046) and a significant increase in PETCO2 ( p= 0.0001) and respiratory rate ( p= 0.041) at Twt in the group of obese children compared with healthy controls; for the second step with the N95 mask, we found no significant difference between the two groups at T30  and T60, while we found a significant increase in PETCO2 (p=0.0062), pulse rate ( p= 0.01) and respiratory rate ( p= 0.0053) at Twt in the group of obese children compared with healthy controls

The authors talk about respiratory function. I agree when they talk about respiratory rate. How does the heart rate fit into "respiratory function" context? The SaO2/CO2 monitoring are expression of gas exchanges rather than respiratory function. I would suggest to consider the change from respiratory function to cardiorespiratory parameters or similar.

Yes, as suggested by reviewer, we have now modified in cardio-respiratory function or cardiorespiratory parameters

Why wasn't spirometry or lung volumes measurement considered? I would suggest to include in the discussion a short "Limitations of the study" paragraph.

Unfortunately, we did not perform spirometry in the children participating in the study, so we have now added the study limitations paragraph. However, we thank the reviewer for the suggestion, in fact we would like in a future work to evaluate also these parameters in obese children wearing the mask

In the abstract, I would change the sentence "obese children who wear a mask are at risk of developing respiratory distress...." as no statistical analysis on risk has been performed, I would suggest to use "are more prone of developing ... (or similar)"

Yes, we have now modified the abstract, using the phrase “overweight and obese children who wear a mask are more prone of developing respiratory distress”

Round 2

Reviewer 1 Report

Thank you for improving your manuscript. However I still feel that you should present the results which you summarize in lines 219 to 227 in a small table. One of the problems with small sample sizes is, that the results often do not reach significant levels. Looking at PETCO2 , the mean  increased from 35  to 39  in healthy children wearing a N95 mask, in overweight or obese children from 37 to 41 which is not such an enormous difference between the 2 groups.

 Lines 164 - 168  and lines 197- 201 are redundant, the detailed statistical results  are included in the table.

 And: there are still some language problems, the manuscript ought to be checked by a native English speaking person ( e.g Table 1 Healthy children, line 149 and 196 same error or table 3: missing "overweight" in the title).

Author Response

Dear Editors and Reviewers,

Thank you for your consideration of our manuscript. We truly think the manuscript is improved after the revisions suggested. Below we respond in detail to the comments and points the reviewer raised. We now submit our revised manuscript for publication in Children. We marked the revisions through the manuscript using the “track changes”.

Reviewer 1

 Thank you for improving your manuscript. However, I still feel that you should present the results which you summarize in lines 219 to 227 in a small table. One of the problems with small sample sizes is, that the results often do not reach significant levels. Looking at PETCO2, the mean  increased from 35  to 39  in healthy children wearing a N95 mask, in overweight or obese children from 37 to 41 which is not such an enormous difference between the 2 groups.

We thank the reviewer for his suggestion. We have now summarized the comparison between two groups in table 4 and in table 5, so that the results are clearer.

Table 4. Comparison between the group of healthy children and the group of children with overweight and obesity wearing a surgical mask. All data were expressed as median, 25th and 75th quartile.

Test with surgical mask

Without mask

T30

With mask

T60

With mask and walking test

Twt

Healthy

Obese

p

Healthy

Obese

p

Healthy

Obese

p

SaO2

%

98

97-98

98

96.75- 98.25

NS

98

97-98

97.5

96.75-98

NS

98

96-98

97.5

96-98

NS

PETCO2

mmHg

36

35-37

37

35.75-39

NS

36

33-37

39

38-43.25

0.0046

35

33-39

42

41-42

0.0001

PR

pulsation/min

96

89-100

93

85.75-100.5

NS

106

100-115

90.5

85.25-102.25

NS

115

110-120

127

118.75-137.25

NS

RR

breaths/min

22

20-23

20

16-21.25

NS

21

21-22

20

16-23

NS

26

24-28

30

25.5-36

0.041

Clinical signs of respiratory distress *

Absent

Absent

Absent

Absent

Absent

Present

Table 5 Comparison between the group of healthy children and the group of children with overweight and obesity wearing an N95 mask. All data were expressed as median, 25th and 75th quartile.

Test with N95 mask

Without mask

T30

With mask

T60

With mask and walking test

Twt

Healthy

Obese

p

Healthy

Obese

p

Healthy

Obese

p

SaO2

%

98

97-98

98

96.75- 98.25

NS

98

96-98

98

97-98

NS

97

96-98

97

96-98

NS

PETCO2

mmHg

35

35-40

37

35.75-39.25

NS

38

35-41

39

38.75-41.75

NS

39

37-41

41.5

40-43.25

0.0062

PR

pulsation/min

99

87-105

93

85.75-100.5

NS

100

82-106

94

88.25-105.25

NS

109

105-114

131

112.25-140

0.01

RR

breaths/min

21

19-22

20

16-21.25

NS

22

20-23

20.5

18-23.25

NS

25

20-30

32

25.5-34

0.0053

Clinical signs of respiratory distress *

Absent

Absent

Absent

Absent

Absent

Present

Lines 164 - 168 and lines 197- 201 are redundant, the detailed statistical results are included in the table.

We agree with the reviewer, we have now deleted lines 164 - 168 and lines 197- 201 and kept only the results in table 2 and table 3.

There are still some language problems, the manuscript ought to be checked by a native English speaking person (e.g Table 1 Healthchildren, line 149 and 196 same error or table 3: missing "overweight" in the title).

We have now corrected the reported errors and checked other grammatical errors through the whole text.